

# Predicting academic performance for students' university: case study from Saint Cloud State University

Bilal I. Al-Ahmad[1,2], Abdullah Alzaqebah[3], Rami Alkhawaldeh[1,4], Ala' M. Al-Zoubi[5], Hsuehi Lo[6] and Adel Ali[2]

[1] Department of Computer Information Systems, Faculty of Information Technology and Systems, The University of Jordan, Aqaba, Jordan
[2] Department of Computing, Informatics, and Data Science, St. Cloud State University, Minnesota, United States
[3] The World Islamic Sciences and Education University, Amman, Jordan
[4] Information Systems Department, Sultan Qaboos University, Muscat, Oman
[5] Department of Data Science and Artificial Intelligence, Faculty of Science and Information Technology, Al-Zaytoonah University of Jordan, Amman, Jordan
[6] College of Education and Learning Design, St. Cloud State University, Minnesota, United States

Corresponding author
Ala' M. Al-Zoubi,
a.alzoubi@zuj.edu.jo

## ABSTRACT

Predicting students' performance is one of the essential educational data mining approaches aimed at observing learning outcomes. Predicting grade point average (GPA) helps to monitor academic performance and assists advisors in identifying students at risk of failure, major changes, or dropout. To enhance prediction performance, this study employs a long short-term memory (LSTM) model using a rich set of academic and demographic features. The dataset, drawn from 29,455 students at Saint Cloud State University (SCSU) over eight years (2016–2024), was carefully preprocessed by eliminating irrelevant and missing data, encoding categorical variables, and normalizing numerical features. Feature importance was determined using a permutation-based method to identify the most impactful variables on term GPA prediction. Furthermore, model hyperparameters, including the number of LSTM layers, units per layer, batch size, learning rate, and activation functions, were fine-tuned using experimental validation with the Adam optimizer and learning rate scheduling. Two experiments were conducted at both the college and department levels. The proposed model outperformed traditional machine learning models such as linear regression (LR), K-nearest neighbor (KNN), decision tree (DT), random forest (RF), and support vector regressor (SVR), and it surpasses two deep learning models, recurrent neural network (RNN) and convolutional neural network (CNN), achieving 9.54 mean absolute percentage error (MAPE), 0.0059 mean absolute error (MAE), 0.0001 root mean square error (RMSE), and an $R^2$ score of 99%.

## INTRODUCTION

Predicting academic performance (*Becker et al., 2014*) is a very essential task in higher education to better assess students outcomes. Higher education institutions grapple with

numerous challenges to foster student success (*McNair et al., 2022*). Consequently, predicting GPA (*Zeineddine, Braendle & Farah, 2021*) is an efficient tool that help to track the progress of students during their study. To that end, universities apply various advanced techniques as machine learning, statistics, and forecasting models to utilize the educational data.

Different educational factors have been investigated to detect the influence on academic success, such as enrollment patterns, student outcomes, teacher evaluations, and gender variance (*Dutt, Ismail & Herawan, 2017*). The studies of *Krumrei-Mancuso et al. (2013)*, *Špakovská et al. (2014)*, *Shieh, Zhifang & Yeh (2014)* investigated motivation as one of the key factors that affect student's success and academic goals. Other factors as psychological factors, study factors, sociological factors have been investigated in the studies of *Špakovská et al. (2014)*, *Shieh, Zhifang & Yeh (2014)*, *Han, Farruggia & Solomon (2022)*. They constructed two models to explore what affects student grades. The first model focuses only on academic training, and the second one captures academic preparation and non-cognitive factors such as motivation and belonging, and they concluded that non-cognitive factors strongly impact student success. Other factors related to the ranking of grades in different colleges, which it sheds light on grade inflation. This issue has been addressed in various studies (*Johnson, 2006*; *Bar, Kadiyali & Zussman, 2009*), suggesting that grade inflation varies among colleges. Grades influence students how to choose the courses, also grading policies helps to distinguish between high-grading and low-grading colleges in universities.

While academic and demographic features play a critical role in term grade point average (GPA) prediction, numerous studies emphasize the impact of non-cognitive factors such as motivation, time management, self-efficacy, emotional well-being, and sense of belonging on academic outcomes as in the studies (*Krumrei-Mancuso et al., 2013*; *Han, Farruggia & Solomon, 2022*). These variables often interact with cognitive factors and institutional contexts to shape student performance. However, collecting non-cognitive and socioeconomic data pose challenges such as reliance on self-reported surveys, concerns over student privacy, and lack of standardization across institutions. In this study, while non-cognitive factors are acknowledged as influential, our dataset is limited to academic and demographic aspects due to the availability of institutional data.

Several studies have applied various types of regression models to predict GPA. For example, the study (*Aydın, 2017*) used multiple linear regression to identify personal factors that affect the prediction of academic success, such as stress, time, and classroom environment. Another study (*Esmat & Pitts, 2020*) applied linear regression to predict the students' success in a science program, to determine the most efficient factors in the core courses. Some studies (*Huberts, Schoonhoven & Does, 2022*; *Krumrei-Mancuso et al., 2013*; *Tinajero et al., 2020*) employed hierarchical regression models to predict student performance; they used Bayesian estimation and Markov chain Monte Carlo methods based on sampling. Similarly, the study (*Krumrei-Mancuso et al., 2013*) built hierarchical linear regression models to predict the success of freshmen through psychological factors. The study (*Tinajero et al., 2020*) utilized hierarchical regression models to predict the

academic success of Spanish students. In their study, they selected perceived social support as an independent factor, and GPA for the first year and third year as dependent variables. Furthermore, the work research (*Hassan, Elkorany & Wassif, 2022*) proposed hybrid model of k-means clustering and regression to predict GPA. The study performed an adjusted R-squared of 0.935 and found that there is positive contribution of time-series clustering with CoI-based feature selection. The study (*Alghamdi & Al-Hattami, 2014*) implemented multiple and logistic regression approach and on 417 Saudi male and female students from three colleges at the University of Dammam. Similarly, the research (*Klomegah, 2007*) applied bivariate and multivariate analytical models on 103 college students studying at a university in North Carolina, the results show that the high school-GPA is a more accurate indicator of a student's academic achievement than the goal-efficacy factors. A comparative study (*Alfadhly, 2024*) has conducted between machine learning and deep learning methodologies, the outcomes found that the linear and bagging regression models are the best predictors for GPA. The work (*Al Madhoun, 2020*) conducted a study in the Gulf Cooperation Council (GCC) that used linear and logistic regression models to predict GPA by testing demographic and academic factors of students.

Other research works employed various machine learning approaches to predict GPA, the studies (*Castro et al., 2007*; *Buenaño-Fernandez, Villegas-CH & Luján-Mora, 2019*; *Alshamaila, Aljarah & Ala'M, 2018*; *Romero & Ventura, 2010*; *Baker & Yacef, 2009*; *Baker, 2010*; *Alangari & Alturki, 2020*; *Maqableh & Alia, 2021*; *Alia, Tamimi & Al-Allaf, 2013*; *Al-Barrak & Al-Razgan, 2016*; *Ala'M, Mora & Faris, 2023*; *Qaddoura et al., 2021*; *Obiedat et al., 2021*; *Aljarah et al., 2021*; *Habib et al., 2018*) applied several prediction models for multiple purposes as data visualization, providing feedback to instructors, recommendations, prediction of student performance, and student modeling. The study (*Alangari & Alturki, 2020*) aims to predict student performance using 15 classification algorithms, the experimental results show both naïve Bayes and Hoeffding tree models achieved the highest accuracy with 91%. The average accuracy for all 15 classifications was about 71%. Another study (*Al-Barrak & Al-Razgan, 2016*) used J48 decision tree to predict GPA based on investigating the grades of prior courses. This work highlighted that the classification rules in mandatory classes could help to evaluate the most important courses in the study plan. Also, the study (*Putpuek et al., 2018*) utilized the C4.5, ID3 decision tree, naïve Bayes, and K-nearest neighbors (KNN) to predict GPA in Rajanagarindra University in Thailand. The authors in *Al-Barrak & Al-Razgan (2016)* have investigated the performance of decision tree-based models. They utilized J48 decision trees to anticipate students' final GPA, highlighting the significance of performance in compulsory courses. According to the findings by *Mohamed et al. (2023)*, ensemble methods, random forest, and gradient boosting have better accuracy when it compared to regression models. The research by *Wang et al. (2015)* proposed to use the SmartGPA model to infer GPA based on analyzing the behavior of students, the results found that there is a positive association between behavioral factors (class attendance, mobility patterns) and academic outcomes. The work in *Ahmed et al. (2021)* introduced the use of supervised learning models for different quarters of GPA at a particular stage. Recent research (*Dewi & Widiastuti, 2020*)

has shown that support vector regression (SVR) yields the best results in predictions based on standardized data, utilizing the radial basis function (RBF) kernel. Furthermore, the research work (*Canagareddy, Subarayadu & Hurbungs, 2019*) used Bayesian and decision tree models to predict the performance of students at the University of Mauritius. They found that the attendance, grades, study time, and health status are more significant indicator to better assess the performance. The study (*Beaulac & Rosenthal, 2019*) implemented random forest to predict students success in the University of Toronto. Similarly, the study (*Panyasai, 2023*) provided an analysis of different models at Rajabhat Rajanagarindra University, and concluded that naïve Bayes was the most effective prediction model.

While numerous studies have applied regression and machine learning models to predict term GPA, the accuracy of these models varies widely across different institutions due to differences in student populations, course structures, grading policies, and data availability. For instance, *Prabowo et al. (2021)* reported a mean absolute error (MAE) of 0.34 and mean square error (MSE) of 0.414 using an multilayer perceptron with long short-term memory (MLP-LSTM) model at Bina Nusantara University, while (*Alnomay, Alfadhly & Alqarni, 2024*) achieved a lower MAE of 0.21 at King Saud University using linear and bagging regression models. Similarly, *Tsiakmaki et al. (2018)* and *Falát & Piscová (2022)* highlighted that models like random forest and decision trees performed better than linear models in their respective institutions. These variations suggest that model performance is sensitive to institutional contexts. Therefore, this study contributes by examining how LSTM can maintain high accuracy within a single institution across both college and department levels, and how it compares favorably to traditional models in similar contexts.

In addition to institutional variation, the performance of prediction models can also differ significantly across departments within the same university. This is often due to variations in curriculum structure, assessment methodologies, and the nature of course content. For example, departments focused on arts and humanities may use qualitative assessments, while engineering departments rely heavily on standardized exams. These differences affect how predictive features relate to GPA and may reduce the model's ability to generalize across academic departments.

The main goal of this article is to predict the academic performance of undergraduate students at the college-level and department-level within a university. In particular, the proposed model purposes to increase the prediction performance of term GPA in the third- year and fourth years. It investigates how the prediction performance of the term GPA is impacted by considering different colleges and departments. The early identification of vulnerable students with low GPA is essential to improve academic advising and counseling (*Thomas, 2002*). Accurate early prediction of term GPA is vital for identifying at-risk students and enabling timely academic interventions. LSTM models are well-suited for this task due to their ability to capture time-dependent patterns in students' academic records. As demonstrated in other fields like hydrology, where LSTM variants have shown strong performance in forecasting complex time-series data (*Waqas & Humphries, 2024*), this model can effectively support early warning systems in education

by detecting potential academic decline before it escalates and will help advisors to identify students at potential risk of academic struggles (*Márquez-Vera et al., 2016*).

## RELATED WORK

Various studies implemented different regression techniques to predict GPA. The research study (*Tsiakmaki et al., 2018*) applied several prediction models using linear regression, support vector machines, decision trees, M5 rules, and k-nearest neighbors. Experiments were built on eight courses, and highlighted that grades in the first semester are a strong predictor of future academic performance. The approach was limited to only eight courses. Another study (*Falát & Piscová, 2022*) built ten predictive models using linear regression, decision trees, and random forest to predict GPA from independent variables; the best accuracy is achieved by a random forest model. Nevertheless, their model is applicable to Faculty of Management and Informatics, University of Zilina, and they used a limited number of sample variables.

In addition, the research work (*Obsie & Adem, 2018*) conducted a comparison among neural networks, linear regression, and SVR prediction models. They found that linear regression and SVR outperformed neural networks, which benefits the usefulness of traditional regression approaches in higher education contexts. Also, the study (*Elbadrawy et al., 2016*) applied two models, course-specific regression (CSpR) and personalized linear multi-regression (PLMR) to predict next-class grade and in-class assessment-prediction in the University of Minnesota. The study is only applicable for College of Science and Engineering. In particular to Bina Nusantara University, the research study (*Prabowo et al., 2021*) proposed a deep learning model (MLP-LSTM) to predict a student's GPA, the study was limited to tabular data. To predict GPA in King Saud University, the authors in *Alnomay, Alfadhly & Alqarni (2024)* applied Linear and bagging regression models, the study was restricted only to College of Engineering and College of Computer. Similarly, the study (*Akuma & Abakpa, 2021*) employed linear regression to predict fourth-year cumulative GPA based on earlier academic records. The approach is only applied to department of Math and Computer Science. This study aims to improve the prediction accuracy of term GPA through adapting more comprehensive analytical approach, which explores all college and department levels. It demonstrates how the prediction performance changes based on respecting the characteristics of each level.

## METHODOLOGY

The proposed methodology aims to predict term GPA for the undergraduate students in college-level and department-level as illustrated in Fig. 1. It investigates academic and demographic features to find the most important features that impact the performance of the prediction model. For the initial preparation step, we removed the students with less than four records to maintain sufficient sequence length for time-series modeling. Sequences of length 2 is utilized to represent at least two consecutive records for each student. For that reason, the proposed model considers at least four records for each single student, two records for training, and

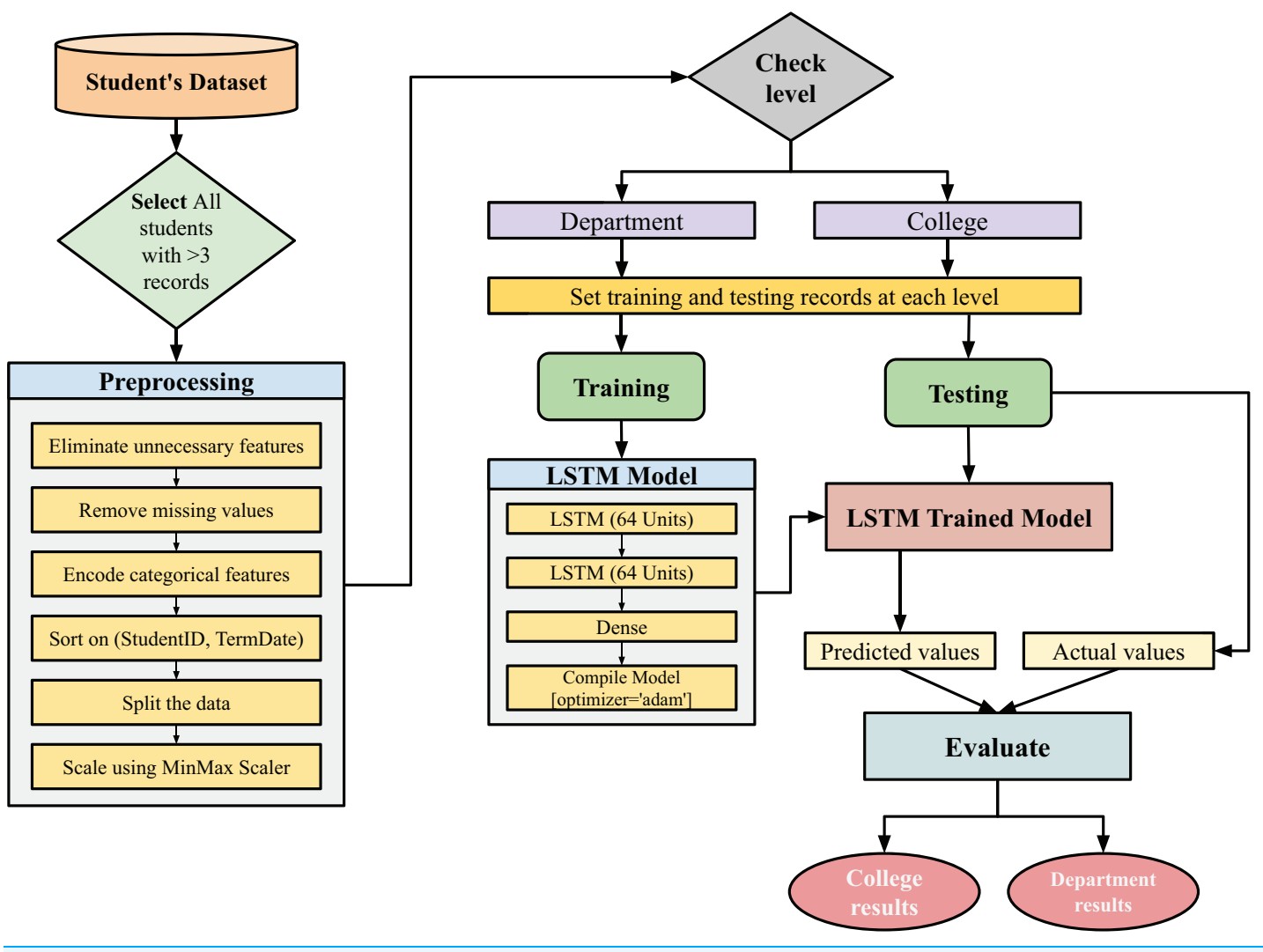

**Figure 1 Flowchart for the proposed model.**

two records for testing. After that, the proposed approach applies the following pre-processing steps.

1. Eliminating irrelevant features to target (term GPA) as (CumulativeLocalCreditsEarned, CumulativeCreditsNotEarned, and CumulativeGPA). These features are related only to Cumulative GPA.

2. Removing all records that have missing values, the number of remaining records is displayed in the Results and Discussions section.

3. Encoding the categorical variables to convert them into a numerical format such as (TermName, ClassCode, TransferStudent, RegularStudent, Gender, Citizenship, Residency, MajorDegree, Department, College, CourseModality, and PriorAssociatesDegree). For instance, Gender has two unique values, "Male" and "Female", the "Male" and "Female" will be converted to 0 and 1. One-hot encoding can

result in a large number of columns. For that, this article uses the label encoding to be more appropriate to ensure that the categorical variables are properly represented for machine learning algorithms.

4. Sorting the dataset based on StudentID and TermDate to filter each single student with all associated terms.

5. Normalizing is an essential step in pre-processing data for machine learning models. It adjusts the range of numerical features to a standard scale, typically [0, 1] or to have a mean of 0 and a standard deviation of 1. This study implements MinMaxScaler from scikit-learn.

**Ethical considerations:** This study was reviewed by the Institutional Review Board (IRB) at St. Cloud State University and was determined to fall outside the purview of the IRB, as it involves only a de-identified dataset of academic records. The research does not involve intervention or interaction with human participants, nor does it contain identifiable private information. As such, IRB approval was not required. The researchers ensured that all data used were fully anonymized to maintain the protection of human subjects.

## Setting training and testing data

After pre-processing, the dataset contains 17 features and 71,277 records for all the investigated colleges and departments. The dataset has at least four records for each single student to utilize the previous terms to predict term GPA better. To enable us to split the data by considering all students who have successfully completed four terms and more. At the college-level, we consider all the students from various departments in the targeted college. For each student, the last two records are split as a testing set and the prior records as a training set. For instance, if the student has seven records, the first five records are considered as training, and the last two records as testing. Besides, at the department-level, all the records for all departments (except the target department) within each college as a training set, and the records for the particular target department as a testing set. After that, the training set will be used to train the LSTM model for predictions, which produces a trained LSTM model that will be used to predict the term GPA in the testing set.

## LSTM model

The impact of the LSTM model has been widely used in natural language modeling, speech, text, machine translation, and other domains. Various studies used the LSTM to accurately predict students' future performance based on historical data since the LSTM can uncover the dependencies and insights that help optimize learning strategies (*Alanya-Beltran, 2024*; *Wan et al., 2023b*; *Wang, 2024*). LSTM is particularly used to capture the temporal dependencies in the academic paths of students from the previous academic terms to the current term.

The proposed approach uses LSTM as an prediction model. The LSTM model facilitates the preservation of information flow (*Febrian et al., 2023*), and mitigate challenges of long term dependencies. In particular, the LSTM model incorporates a series of recurrent module which is considered as a specialized variant of recurrent neural network (RNN).

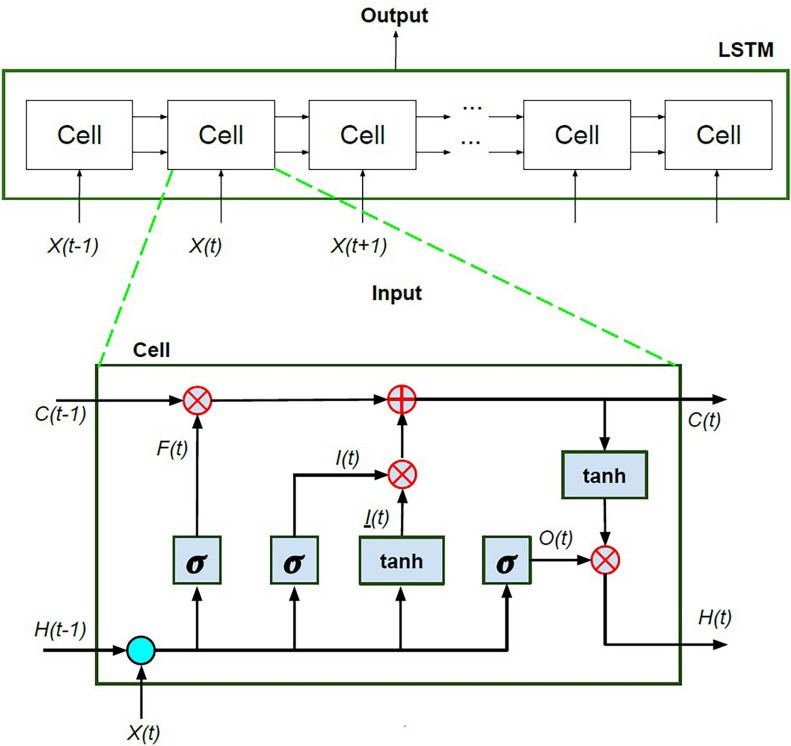

**Figure 2 The components of LSTM architecture.**

The LSTM model propagates information through connection links that is inherently dependent on temporal steps. It is technically implemented through a recurrent cell structure, where individual cells are interconnected across time steps, as illustrated in Fig. 2. This mechanism handles a stable cell state and maintain the reliability preservation of gradient based information, thereby mitigating the risk of information degradation over time.

LSTM is the best-fit model for this study because the dataset consists of short sequential academic records (2–6 terms per student) over 8 years, making temporal modeling essential (*Prottasha et al., 2022*). LSTM models are known for handling short- to medium-range dependencies as indicated in the studies (*Yousafzai et al., 2021*; *Hochreiter, 1997*), and they have been widely used in educational contexts for modeling student progress over time (*Piech et al., 2015*). With over 71,000 records from 29,455 students, the data size is sufficient for LSTM but not ideal for more complex models like Transformers. Transformer-based models are powerful, however because of their complexity and extensive number of parameters, they typically need a lot more data to train efficiently (*Wu et al., 2019*, *2021*; *Baz, 2024*). In contrast, LSTMs (*Shakor & Khaleel, 2025*; *Ghazvini, Sharef & Sidi, 2024*) are more data-efficient and simpler to train, and it has sequential learning power which improves feature extraction. Thus, it has superior ability to optimize the evolving academic patterns of students.

As displayed in Fig. 2, the LSTM model is attributed by two main parts: the hidden state $H(t)$, which is dynamically updated over time, and the cell state $C(t)$, which captures

long-term memory. The cell state $C(t)$ is modulated along the top horizontal line of the LSTM cell, where information is added or removed during a series of gating mechanism. The behavior of each gating element is controlled by learning parameters or weights, which optimized during the training phase. The LSTM cell consists of three essential gates: forget gate, input gate, and output gate. The forget gate $F(t)$ regulates the influence of the input $X(t)$ and the previous hidden state $H(t-1)$ on the cell state $C(t)$. This gate decides whether to retain or discard information from $X(t)$, with $H(t-1)$ being checked on the binary output of a sigmoid activation function. The input gate, consisting $I(t)$ and $I(t)$, controls the degree to which the input value is integrated into the cell state $C(t)$. The output gate $O(t)$ is responsible for generating predictions at each time step by determining the final output by considering the current cell state $C(t)$. The operations associated with these gates are mathematically formalized in Eqs. (1)–(6).

Furthermore, the LSTM model uses four fundamental functions: the sigmoid ($\sigma$), which produces gate activation values; hyperbolic tangent function ($tanh$), which scales values to a normalized range, and multiplication ($\times$) and summation ($+$), which are used to combine and transform data. These functions collectively enable efficient weight adjustments during the back propagation process. Where $W$ and $B$ represent the weight matrix and bias vector. The sigmoid function is expressed as $sigma(cdot)$, and the hyperbolic tangent function is denoted by $tanh(cdot)$ (*Jiang et al., 2021*). The weights associated with the forget gate, input gate, and output gate are represented as $W_f$, $W_i$, and $W_o$, respectively. The LSTM layers capture sequential patterns while handling the issue of vanishing gradients (*Sangiorgio & Dercole, 2020*). The weights across the entire network, LSTM layers throughout back propagation and gradient descent, are iteratively updated to optimize model performance.

$$F(t) = \sigma(W_f[H(t-1), X(t)] + B_f) \tag{1}$$

$$I(t) = \sigma(W_i[H(t-1), X(t)] + B_i) \tag{2}$$

$$O(t) = \sigma(W_o[H(t-1), X(t)] + B_o) \tag{3}$$

$$\underline{I}(t) = \tanh(W_i[H(t-1), X(t)] + B_i) \tag{4}$$

$$C(t) = F(t) \cdot C(t-1) + I(t) \cdot \underline{I}(t) \tag{5}$$

$$H(t) = O(t) \cdot \tanh(c(t)) \quad \left( \tanh(x) = \frac{\sinh}{\cosh} = \frac{e^x - e^{-1}}{e^x + e^{-1}} \right). \tag{6}$$

The architecture of the LSTM consists of two layers, each containing 64 units, followed by dense output layer. The number of units in each layer defines the dimensionality of the output space. The first layer is designed to capture short term dependencies between elements within the sequence, while the second layer identifies long term temporal dependencies from extended sequences. Once the LSTM layers have extracted temporal features from the input sequences, a dense layer is employed to map these features into the output space for final prediction. After that, during the compilation step, the optimizer and evaluation metrics are specified. The proposed approach utilizes Adam optimizer (*Arya & Hanumat Sastry, 2022*), which improves the model performance by adaptively adjusting

the learning rate throughout the training phase. Finally, the predicted values for college-level and department-level are compared to the actual values during the evaluation step to assess the accuracy of the model.

## Evaluation metrics

This study deploys the evaluation metrics to assess the model's performance (*Mansour, Obeidat & Hawashin, 2023*; *Kanan et al., 2023*, *2022*). The metrics of mean absolute error (MAE), mean square error (MSE), root mean square error (RMSE), $R^2$, and mean absolutel percentage error (MAPE) expressed in Eqs. (7)–(11).

1. MAE: is a measurement that calculates the average size of the errors in a group of predictions, regardless of their direction.

$$\text{MAE} = \sum_{i=1}^{D} |x_i - y_i|. \tag{7}$$

2. MSE: it calculated as the difference between predicted ($\hat{y}$) and actual (y) values.

$$\text{MSE}(y, \hat{y}) = \frac{\sum_{i=0}^{N-1} (y_i - \hat{y}_i)^2}{N}. \tag{8}$$

3. RMSE is a frequently used metric for assessing the precision of a model in regression analysis. RMSE is the square root of the average of the squared differences between predicted and actual values.

$$\text{RMSE}(y, \hat{y}) = \sqrt{\frac{\sum_{i=0}^{N-1} (y_i - \hat{y}_i)^2}{N}}. \tag{9}$$

4. R-squared ($R^2$) assesses the accuracy of a regression model by showing how much variation in the outcome variable can be explained by the explanatory variables.

$$R^2(y, \hat{y}) = 1 - \frac{\sum_{i=1}^{N} (y_i - \hat{y}_i)^2}{\sum_{i=1}^{N} (y_i - \bar{y})^2}. \tag{10}$$

5. Mean Absolute Percentage Error (MAPE): measures the accuracy of the forecasting model, it calculates the average percentage difference between predicted and actual values, where a lower MAPE signifies higher prediction accuracy.

$$\text{MAPE}(y, \hat{y}) = \frac{100\%}{N} \sum_{i=0}^{N-1} \frac{|y_i - \hat{y}_i|}{|y_i|}. \tag{11}$$

## RESULTS AND DISCUSSIONS

This section contains description about dataset, features importance, experimental setting, college-level results, department-level results, and the validation for the proposed approach.

## Dataset description

In this experiment, we collected a real dataset of undergraduate students from a public 4-year institution, namely Saint Cloud State University, for a time of 8 years (2016–2024).

**Table 1 Description of SCSU dataset.**

| Feature number | Feature name | Description |
|---|---|---|
| 1 | RandomId | A unique identifier for each student |
| 2 | TermName | The term during which the data was recorded |
| 3 | ClassCode | The student's class (*e.g.*, SR for senior, JR for junior) |
| 4 | TransferStudent | Indicates if the student is a transfer student (Y/N) |
| 5 | RegularStudent | Indicates if the student is a regular student (Y/N) |
| 6 | Gender | Gender of the student (Male/Female) |
| 7 | Citizenship | Citizenship status of the student |
| 8 | Residency | Residency status (In-State/Out-of-State) |
| 9 | MajorDegree | The major and degree of the student |
| 10 | Department | The department offering the major |
| 11 | College | The college within the university |
| 12 | PriorAssociatesDegree | Indicates if the student has a prior associate's degree |
| 13 | CourseModality | The modality of the courses taken |
| 14 | TermEnrolledCredits | Number of credits enrolled in the term |
| 15 | TermGPA | GPA for the particular term (Fall, Spring, Summer) |
| 16 | TermLocalCreditsEarned | Local credits earned during the term |
| 17 | TermCreditsNotEarned | Credits not earned during the term. |
| 18 | CumulativeLocalCreditsEarned | Cumulative local credits earned |
| 19 | CululativeCreditsNotEarned | Cumulative credits not earned |
| 20 | CumulativeGPA | Cumulative GPA |

**Table 2 Statistics about each academic college.**

| College | No. records | No. departments | No. majors/degrees | No. students |
|---|---|---|---|---|
| Education and Learning Design | 8,012 | 5 | 12 | 2,171 |
| Health and Wellness Professions | 25,995 | 8 | 26 | 7,395 |
| Science and Engineering | 29,646 | 10 | 68 | 7,829 |
| Liberal Arts | 19,563 | 15 | 48 | 5,553 |
| Herberger Business | 24,768 | 8 | 22 | 6,527 |

This data set contains 108,336 records of anonymous students with 20 characteristics for 29,455 students. Each record represents a single term for a particular student. The data set contains various features related to student demographics, academic performance, and enrollment details. Table 1 shows the description for all features. The data set contains five academic colleges and 46 academic departments. Table 2 shows statistics about these colleges and departments.

## Experimental setting

The proposed method is developed using the Python programming language with the aid of tensorflow, keras, sklearn, pandas, and numpy packages, on an Intel(R) Core i7 CPU operating at 2.00 GHz with 16 GB of RAM. Table 3 shows the parameter setting in this research. The LSTM model consists of two layers with 64 units each, followed by a dense

**Table 3 The parameters setting used in this experiment.**

| Parameter | Value |
| --- | --- |
| LSTM layers | 2 (64 units for each) |
| Dense layers | 1 (1 unit) |
| Optimizer | Adam optimizer (learning rate of 0.001) |
| Activation function | ReLU |
| Loss function | Mean squared error |
| Batch size | 64 |
| epochs | 100 |
| Validation split | 20% |
| Learning rate adjustment | ReduceLROnPlateau |

output layer. It was trained for 100 epochs with a batch size of 64, using the Adam optimizer with learning rate = 0.001 and ReLU activation. The MSE loss function was used, and 20% of the training data was set aside for validation. To prevent overfitting and improve convergence, a ReduceLROnPlateau callback was applied.

In addition, the experimental settings used various parameters. First, a sequence length of 2 was chosen to allow the LSTM model to capture short-term temporal patterns between consecutive academic terms (*Brownlee, 2017*; *Graves & Graves, 2012*). This length balances the need for time-series learning with data availability across all students. Also, using a length of 2 ensures consistent, reliable inputs and supports accurate early prediction without overfitting. Second, the label encoding was used instead of one-hot encoding to reduce feature dimensionality and avoid sparsity, which can negatively impact LSTM performance. One-hot encoding would have significantly increased the input size due to many categorical variables. Label encoding, combined with normalization, offers a more efficient representation while maintaining model stability and performance in a sequence-based context. Third, MinMaxScaler was utilized because it scales data to [0, 1], which aligns well with LSTM's use of sigmoid and tanh activation functions, enhancing training stability and convergence. MinMaxScaler helps prevent vanishing gradients. It was also preferred over RobustScaler since the dataset was pre-cleaned with minimal outliers. This choice improves model performance by ensuring inputs remain within an optimal range for time-series learning.

To further support the rationale behind these architectural settings, the experimental settings were selected based on theoretical considerations and insights from recent literature in time series modeling (*Waqas, Humphries & Hlaing, 2024*; *Alanya-Beltran, 2024*; *Wan et al., 2023a*; *Landi et al., 2021*; *Waqas et al., 2024*; *Baz, 2023*). The two layers design balances model complexity and generalization, and it allows to capture both short-term and long-term academic data without overfitting. The use of 64 units per layer aligns with established practices in sequential data modeling (*Jiang et al., 2021*), this dimensionality supports sufficient learning capacity while preserving computational efficiency. ReLU was chosen as the activation function for its ability to mitigate vanishing gradient, and Adam optimizer with learning rate of 0.001 is widely used as robust for

adaptive learning in LSTM context (*Arya & Hanumat Sastry, 2022*). These hyperparameters were determined through empirical tuning to optimize prediction accuracy and training stability.

### Features importance

Feature importance involves calculating the score for all input variables to identify significant features through prediction. The higher feature score, the larger impact on the prediction model. Under the permutation method (*Huang, Lu & Xu, 2016*; *Altmann et al., 2010*), the feature importance is calculated by noticing the increase or decrease in error when we permute the values of a feature. If permuting the values causes a huge change in the error, it means the feature is important for our model. Figure 3 shows the correlation coefficient between different features and term GPA.

The feature importance results show that academic variables (TermCreditsNotEarned, LocalCreditsEarned, and TermEnrolledCredits) had the strongest impact on term GPA prediction, while demographic and external factors (Gender, Residency) had lower predictive weight. This indicates that academic engagement plays a more direct role in student performance than static personal attributes. Such features, TermCreditsNotEarned, LocalCreditsEarned, and TermEnrolledCredits, directly measure students' academic engagement and performance within a given term and dynamically linked with with academic behavior patterns that the LSTM model is designed to capture. In contrast, Gender and Residency are static demographic attributes that do not change over the time or reflect actual academic activity as they only indirectly related to academic performance, which limits their predictive power in a time-series model like LSTM. Including these demographic features still enhances transparency by allowing the model to assess their influence fairly, and their lower importance supports the model's focus on actionable academic indicators, reinforcing both validity and explanation of the findings.

Features as (CourseModality and TransferStudent) were included as input features to enhance model robustness. Course modality influences academic performances due to differences in instructional delivery, students' participation, and access to resources. Including this feature enable the model to adjust to various learning environments. Likewise, Transfer students are a critical demographic factor, as they often lack to have a consistent academic history and encounter transnational challenges as transfer credits issues, which can impact term GPA. To handle this, the model uses available academic records while normalization and sequence filtering ensures that only students with at least four academic records are included. These features improve the generalizability of the prediction model through capturing diverse academic pathways and institutional experiences.

As term GPA relies on number of credit hours, the TermCreditsNotEarned represents a strong positive correlation with GPA prediction. The reason is when students have more remaining credit hours tends to provide the prediction model with more academic records which leads to obtain a higher prediction performance. Similarly, The TermLocalCreditsEarned shows when students earned more local credits, the LSTM model can perform higher prediction. The TermLocalCreditsEarned impact the prediction

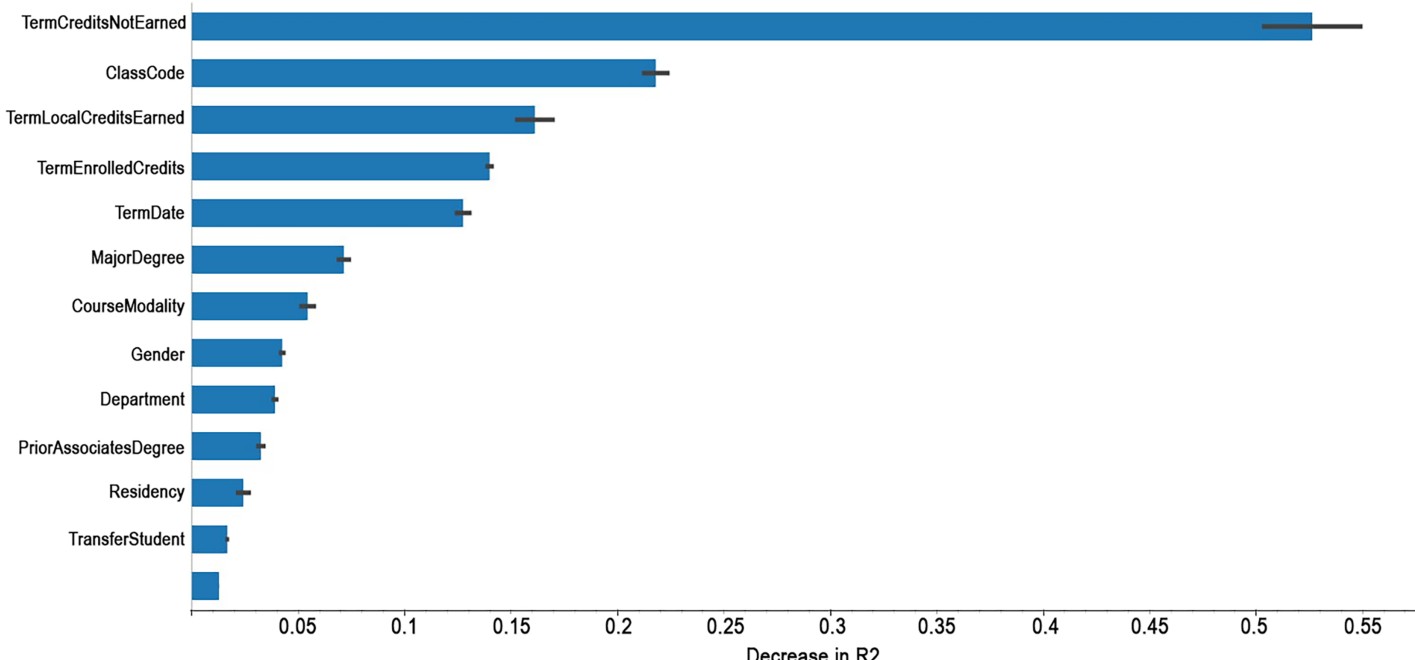

**Figure 3 The importance of features for term GPA.**

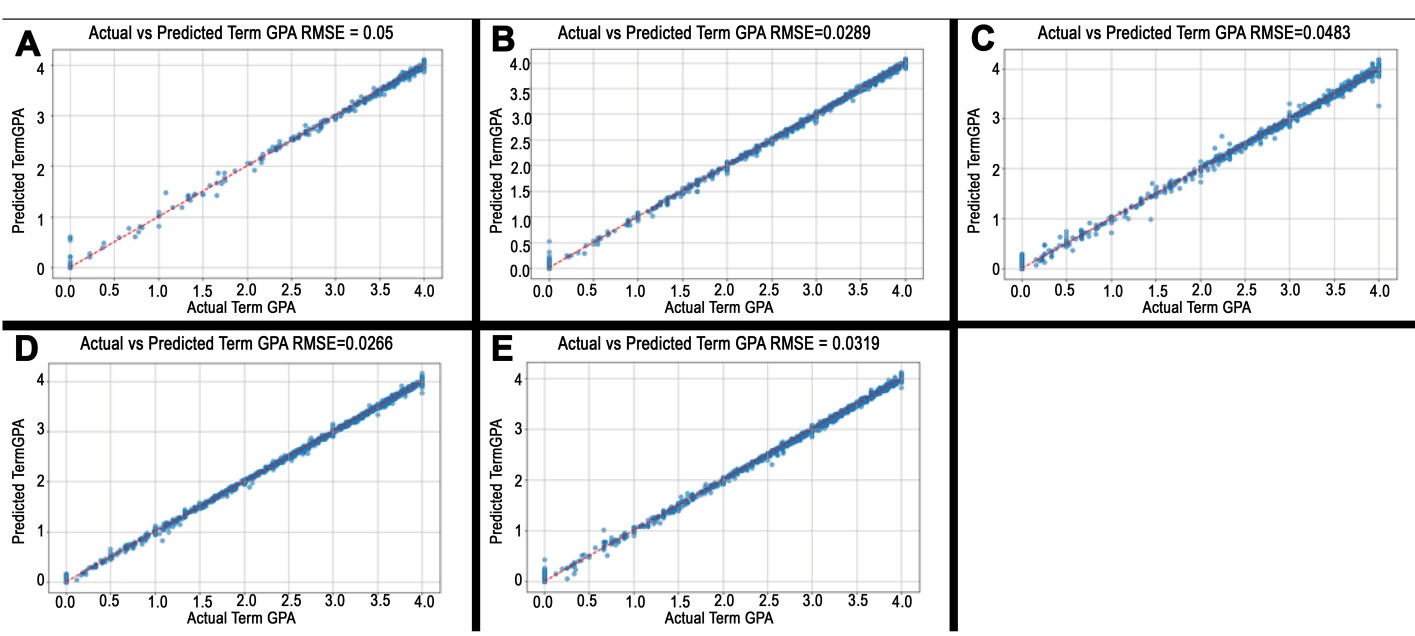

**Figure 4 Predicted *vs.* actual values plots.** (A)–(E) illustrate the model's performance in predicting student GPA across five colleges. In each subfigure, the x-axis represents the actual Term GPA, while the y-axis represents the predicted Term GPA. The distribution of points reflects the accuracy of the prediction within each college. (A) represents the model's performance over the College of Education and Learning Design. At the same time, (B) through (E) illustrate the model's performance over the College of Health and Wellness Professions, College of Liberal Arts, College of Science and Engineering, and Herberger Business College, respectively.

model. In Fig. 4, it shows students with low term GPA in College of Education and Learning Design (CoELD) has difference between actual and predicted values when it compared to College of Science and Engineering (CoSE). The CoELD adapts performance driven assessment while the CoSE is content driven assessment. Courses in the CoELD have training pedagogy and constructed based on different teaching and assessment styles. Variance in assessment makes effect on prediction performance. Furthermore, the TermEnrolledCredits shows the higher enrolled credits in a particular academic term which tends to have higher prediction results.

Switching course modality influences outcomes and assessment strategies, during COVID-19 pandemic, SCSU developed different course modality, in-person, on-line, and hybrid. Some students prefer to take online modality rather than in-person, this may causes changes in term GPA. Normally, transfer students are moving from a 2-year community college to new structure of 4-year university. The transfer students may show low term GPA in the first and second terms because the learning environment is different. As the proposed model mainly focuses on the prediction for third year and fourth year of study, transfer students do not have previous term GPA data as regular students in SCSU. As result, this would impact the prediction accuracy.

It is worth noting that there are certain influential factors such as course difficulty, class size, instructor effects, socioeconomic variables were excluded in this study. The primary reason is the unavailability of such features due to the university policy and regulations within the investigated dataset. Further, integrating behavioral or contextual features often requires cross-system access, which raises privacy and consistency challenges.

## College-level results

The results represent that students majoring in different colleges have different term GPA. Standing on the educational perspectives, this study provides a justification of why the results shows slightly different prediction accuracy.

Based on RMSE value, Table 4 shows the highest, medium, and the lowest predication accuracy in colleges are Science and Engineering, Herberger Business, and Education and Learning Design, respectively. The variations in prediction accuracy can be explained by the differences in the courses offered and assessment styles by these colleges (*Sin & Soares, 2020*). The College of Science and Engineering, traditionally provides content-driven courses, where the assessments are based on technical measurable outputs as conducting labs, coding, experiments, and design solutions. The Herberger Business College offers completion-driven course, where assessments are constructed based on applying knowledge to real world applications, scenarios, code (*Al-Ahmad et al., 2023*), projects, and case studies. On other hand, the College of Education and Learning Design has communication-driven courses, where the assessments are mostly subjective, construed on respecting multiple teaching styles and interpersonal skills between teachers and different learners as adults, children, or special needed students. For example, in the classroom management course, instructors teach social emotional learners and regular learners. Consequently, assessment of such course is different from lab or experiment test as conducted in the college of Science and Engineering. These factors introduces

**Table 4 A comparison results for different colleges.**

| College | MAE | MSE | RMSE | $R^2$ | MAPE |
|---|---|---|---|---|---|
| College of Education and Learning Design | 0.0247 | 0.0025 | 0.0500 | 0.9961 | 1.20E+13 |
| College of Health and Wellness Professions | 0.0180 | 0.0008 | 0.0289 | 0.9990 | 8.11E+12 |
| College of Liberal Arts | 0.0265 | 0.0023 | 0.0483 | 0.9978 | 1.20E+13 |
| College of Science and Engineering | 0.0179 | 0.0007 | 0.0266 | 0.9993 | 5.76E+12 |
| Herberger Business College | 0.0194 | 0.0010 | 0.0319 | 0.9987 | 7.47E+12 |

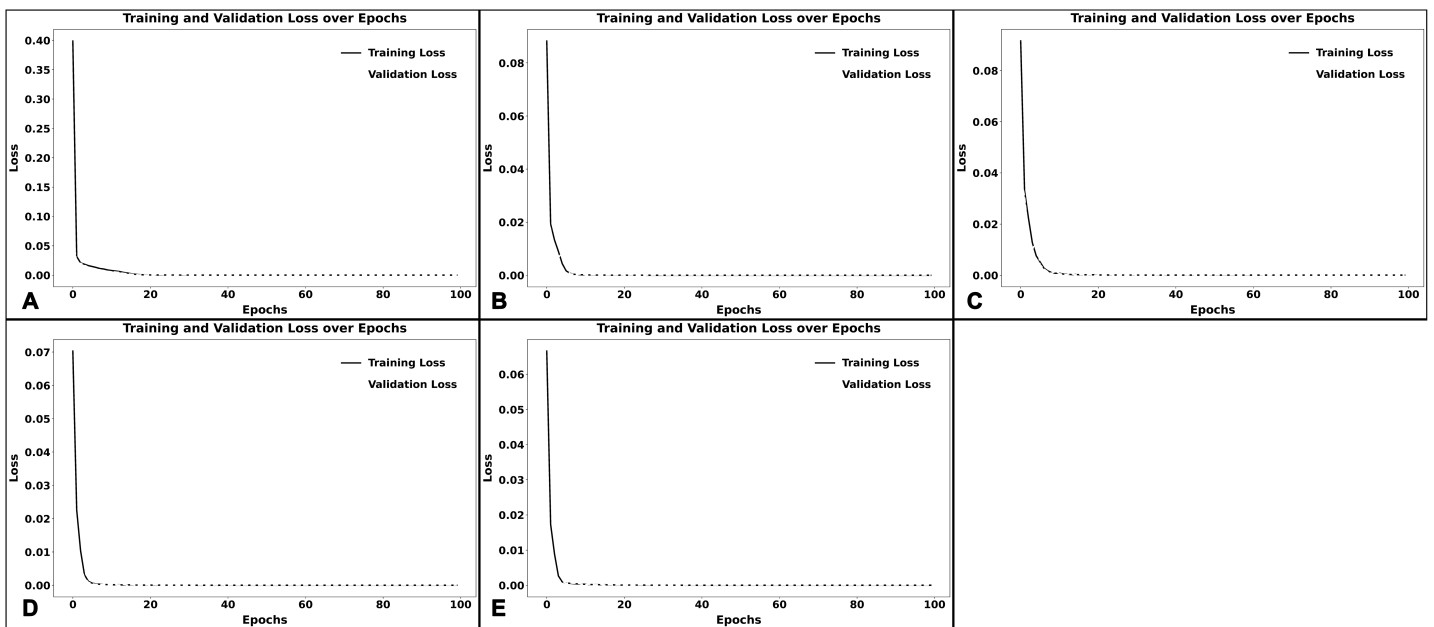

**Figure 5 Learning curves.** (A)–(E) illustrate the learning curves of the model when trained on data from each college. (A) represents the learning curve of the College of Education and Learning Design. At the same time, (B) through (E) illustrate the learning curves of the College of Health and Wellness Professions, College of Liberal Arts, College of Science and Engineering, and Herberger Business College, respectively. Each plot displays how training and validation loss evolved over epochs, helping assess model performance and potential overfitting. A smaller gap between the two curves indicates better generalization.

inconsistencies that challenges the prediction model. Obviously, subjective assessment is harder than objective assessment and this would make a variance of the prediction performance of term GPA.

Figure 4 illustrates the scatter plots of predicted *vs.* actual term GPA values in all investigated colleges over the test set, showing the achieved RMSE values at the top of each sub-figure. The differences are reported as 5%, 2.8%, 4.8%, 2.6%, and 3.2%, for College of Education and Learning Design, College of Health and Wellness Professions, College of Liberal Arts, College of Science and Engineering, and Herberger Business College, respectively. College of Science and Engineering highlights most accurate prediction and College of Education reflects least accurate performance. While most colleges exhibit relatively small differences, which indicate strong alignment between predicted and actual term GPA values, slight variability exist depending on characteristics of each college.

In addition, Fig. 5 represents the model's learning curves with the loss values compared to training epochs for both the training and validation sets, indicating a normal training behavior without overfitting in the proposed model. It is clearly shown that the validation loss roughly corresponds to the training loss for the College of Education and Learning, College of Science and Engineering, and Herberger Business College, with no notable divergence, indicating that the model is not overfitting and can generalize well to new data. On the other hand, and for the College of Health and Wellness Professions and the College of Liberal Arts, there is a small divergence during the early epochs, when the validation loss marginally increases while the training loss reduces. This early gap is likely a result of an infrequent overfitting phase in which the model attempts to memorize the training data before completely capturing generalizable patterns.

The discrepancy in prediction accuracy across different colleges stems from differences in evaluation styles and curriculum structure. For example, College of Science and Engineering tends to use objective, quantifiable assessment such as exams, lab reports, and problem solving tasks, which result in more consistent term GPA patterns and higher model accuracy. In contrast, College of Education and Learning Design rely on subjective based evaluation, such as classroom engagement and teaching simulations, which lead to have more variability and make it harder to predict term GPA. These differences impact how well LSTM model can learn and generalize patterns.

## Department-level results

Evaluation of students depends on certain factors as time management (*Wolters & Brady, 2021*), students motivation (*Spurlock, 2023*), students roles in homogeneous or heterogeneous groups (*Briggs, 2020*). Also, different university may have different academic structures (*Delbanco, 2023*) as courses, program maps, or majors. If the structure is harder, the low-expectation students encounter some challenges that increases possibilities of failure or having low term GPA. This study aims to explore why there are prediction variations among departments in the same college. These differences can be attributed to the nature of courses and assessment methods within each department.

As shown in Table 5, the highest, medium, and lowest prediction accuracy of departments are shown in Information Media, Special Education, and Child and Family Studies, accordingly. The courses in Information Media concentrating on how to use e-Learning and online media tools to deliver content knowledge to K-12 students or special-needed students. The assessments are structured on integrating technology to improve the learning process. The technology and relatively objective evaluations contributes to higher prediction accuracy of term GPA. The courses in Special Education department are designed to address different types of special needed students. Even though rich theories are taught in special education courses, but the assessments are still based on how much improvements achieved by special education students compared to the normal students. The variability in learning among special needed students indicates to find moderate prediction performance. The courses from the Child and Family Studies department provide students to learn how to develop the creativity of children and family metrics. Comparing to the previous two departments, it is reasonable to observe that the

**Table 5  A comparison results for departments in College of Education and Learning Design.**

| Department | MAE | MSE | RMSE | R² | MAPE |
|---|---|---|---|---|---|
| Child and Family Studies | 0.1187 | 0.0212 | 0.1456 | 0.9528 | 1.53123E+13 |
| Information Media | 0.0220 | 0.0010 | 0.0312 | 0.9979 | 3.12119E+12 |
| Social Sciences | 0.0389 | 0.0027 | 0.0523 | 0.9966 | 1.128E+13 |
| Special Education | 0.0383 | 0.0038 | 0.0615 | 0.9917 | 1.34589E+13 |
| Teacher Development | 0.0661 | 0.0087 | 0.0931 | 0.9800 | 7.21313E+12 |

**Table 6  A comparison results for departments in College of Health and Wellness Professions.**

| Department | MAE | MSE | RMSE | R² | MAPE |
|---|---|---|---|---|---|
| Communication Sciences and Disorders | 0.0145 | 0.0003 | 0.0183 | 0.9989 | 2.98786E+11 |
| Community Psychology and Counseling Family | 0.0188 | 0.0012 | 0.0344 | 0.9986 | 6.33853E+12 |
| Criminal Justice | 0.0257 | 0.0012 | 0.0345 | 0.9986 | 4.1742E+12 |
| Kinesiology | 0.0212 | 0.0019 | 0.0433 | 0.9968 | 3.10335E+12 |
| Medical Laboratory Science | 0.0086 | 0.0001 | 0.0121 | 0.9997 | 1.34394E+12 |
| Nursing | 0.0185 | 0.0006 | 0.0235 | 0.9987 | 9.53822E+11 |
| Social Work | 0.0161 | 0.0004 | 0.0212 | 0.9991 | 1.70518E+12 |

prediction performance is slightly lower in this department as it conducts qualitative assessment, which results in challenges to have consistent evaluations.

Referring to College of Health and Wellness results displayed in Table 6, the highest, medium, and lowest prediction accuracy of departments are observed in Medical Laboratory Science, Nursing, and Kinesiology, respectively. The courses from Medical Laboratory Science department emphasize applying scientific process to evaluate students' understanding. Assessment in this department are more objective and rely on standard metrics such as labs procedures, diagnostics accuracy, and medical policies. Such assessments reflect that this department achieves the highest prediction. In the Nursing department, courses focuses more on behavioral and interpersonal aspects, such as understanding human feelings and interactions. Assessments are designed to respect outcomes related to hospital procedures and treatment with patients. The combination of behavioral evaluation and procedural metrics results in medium prediction performance. Courses in the Kinesiology departments incorporate a broader scope of content knowledge, societal awareness, and human characteristics. Assessments in this department often include multiple dimensions of physical environment, human body, and psychological components. The interdisciplinary nature of these courses combined with subjective evaluation, leads to the lowest prediction.

With regard to College of Liberal Arts, as indicated in Table 7, the highest, medium, and lowest prediction accuracy of departments are occurred in Language and Culture, Theater and Film Studies, Anthropology, correspondingly. Courses in the Language and Culture department focus on teaching various languages and promoting cultural understanding (*Altarriba & Basnight-Brown, 2022*), language reflects culture; as students gain proficiency in language, they develop a deeper understanding of the related culture. This cultural

**Table 7 A comparison results for departments in College of Liberal Arts.**

| Department | MAE | MSE | RMSE | $R^2$ | MAPE |
|---|---|---|---|---|---|
| Anthropology | 0.0632 | 0.0093 | 0.0967 | 0.9916 | 3.70177E+13 |
| Art | 0.0271 | 0.0035 | 0.0595 | 0.9938 | 5.83334E+11 |
| Communication Studies | 0.0308 | 0.0023 | 0.0478 | 0.9972 | 1.07428E+13 |
| English | 0.0229 | 0.0011 | 0.0336 | 0.9985 | 3.36119E+12 |
| Ethnic, Gender and Women Studies | 0.0293 | 0.0026 | 0.0514 | 0.9979 | 1.25261E+13 |
| Global Studies | 0.0125 | 0.0003 | 0.0159 | 0.9992 | 1.18789E+12 |
| History | 0.0135 | 0.0004 | 0.0206 | 0.9996 | 3.07661E+12 |
| Languages and Cultures | 0.0082 | 0.0001 | 0.0114 | 0.9999 | 2.19226E+12 |
| Mass Communication | 0.0150 | 0.0004 | 0.0203 | 0.9994 | 1.20737E+12 |
| Music | 0.0126 | 0.0003 | 0.0160 | 0.9996 | 1.36426E+12 |
| Philosophy | 0.0135 | 0.0008 | 0.0282 | 0.9990 | 1.07631E+12y |
| Political Science | 0.0108 | 0.0002 | 0.0153 | 0.9997 | 3.91267E+12 |
| Psychology | 0.0134 | 0.0003 | 0.0182 | 0.9996 | 1.95155E+12 |
| Sociology | 0.0112 | 0.0003 | 0.0163 | 0.9998 | 2.10004E+12 |
| Theatre and Film Studies | 0.0137 | 0.0004 | 0.0200 | 0.9993 | 6.6189E+11 |

**Table 8 A comparison results for departments in College of Science and Engineering.**

| Department | MAE | MSE | RMSE | $R^2$ | MAPE |
|---|---|---|---|---|---|
| Atmospheric and Hydrologic Sciences | 0.0103 | 0.0002 | 0.0140 | 0.9997 | 1.23588E+12 |
| Biological Sciences | 0.0582 | 0.0066 | 0.0813 | 0.9915 | 3.80526E+12 |
| Chemistry and Biochemistry | 0.0153 | 0.0005 | 0.0217 | 0.9995 | 3.07345E+12 |
| Computer Science Information Technology | 0.0130 | 0.0006 | 0.0237 | 0.9993 | 2.91149E+12 |
| Electrical and Computer Engineering | 0.0117 | 0.0003 | 0.0165 | 0.9997 | 1.36516E+12 |
| Environmental and Technological Studies | 0.0115 | 0.0003 | 0.0168 | 0.9996 | 5.54411E+11 |
| Geography and Land Surveying | 0.0135 | 0.0003 | 0.0164 | 0.9995 | 4.88741E+11 |
| Mathematics and Statistics | 0.0081 | 0.0002 | 0.0128 | 0.9998 | 4.6298E+11 |
| Mechanical and Manufacturing Engineering | 0.0097 | 0.0002 | 0.0131 | 0.9998 | 1.60119E+12 |
| Physics and Astronomy | 0.0071 | 0.0001 | 0.0098 | 0.9999 | 2.16852E+11 |

intuition improves their ability to perform better in subsequent terms. Assessments in this departments include mix of objective and subjective methods that make balance grading system, this would help the proposed model to achieve higher accuracy as students improve their linguistic and cultural skills over the time. In the Theater and Film Studies department, courses mainly helps students develop their creative ability in theater and film production. Assessment of such courses are basically subjective metrics. Such subjectivity often leads to variability in evaluations, as students and instructors may have different perspectives and artistic criteria, resulting in moderate prediction. Anthropology courses aim to explore theories of explaining the past and future of human culture and activities. Assessment methods in such discipline are harder to precisely reflect student's outcomes, which shows lower prediction performance.

**Table 9 A comparison results for departments in the Herberger Business College.**

| Department | MAE | MSE | RMSE | $R^2$ | MAPE |
|---|---|---|---|---|---|
| Accounting | 0.0243 | 0.0016 | 0.0395 | 0.9976 | 8.50174E+12 |
| Economics | 0.0173 | 0.0005 | 0.0233 | 0.9993 | 5.15932E+12 |
| Finance, Insurance and Real Estate | 0.0145 | 0.0005 | 0.0229 | 0.9992 | 1.98643E+12 |
| Hospitality and Tourism | 0.0123 | 0.0003 | 0.0179 | 0.9995 | 1.12389E+11 |
| Information Systems | 0.0098 | 0.0002 | 0.0144 | 0.9997 | 7.53558E+11 |
| Management and Entrepreneurship | 0.0133 | 0.0003 | 0.0181 | 0.9994 | 2.22306E+12 |
| Marketing | 0.0112 | 0.0002 | 0.0155 | 0.9996 | 1.18789E+12 |
| Planning and Community Development | 0.0127 | 0.0003 | 0.0163 | 0.9997 | 4.42879E+12 |

In the College of Science and Engineering as represented in Table 8, the prediction accuracy varies across departments. The Physics and Astronomy department reveals the highest prediction values, courses are reflecting strongly consistent and well defined structure. Assessments are mainly objective and they based on theoretical understanding and analytical problem solving skills, leading to better prediction performance. The Chemistry and Biochemistry department demonstrates moderate prediction performance, evaluations are combination of objective and subjective, such as lab experiments and applied instructional methods. Biological Science department shows the lowest prediction value, assessments are primarily subjective and depend on qualitative clinical metrics. Courses are designed as preliminary or prerequisites for medical field as nursing and medical labs, which impacts overall prediction reliability.

The results of Herberger Business College, as given in Table 9, the highest prediction accuracy is observed in the Information Systems department, followed by moderate accuracy in the Management and Entrepreneurship department, and the lowest in Accounting department. Courses in Information Systems department emphasize practical knowledge, coding skills, and software development. The technical based assessment contributes to higher accuracy, as outcomes are more measurable and consistent. Courses in Management and Entrepreneurship are business driven and developed around abstract business concepts as leadership, innovation, and strategic management. The conceptual based assessments presents moderate prediction. The accounting department focuses on content intensive courses that include advanced statistical methods, real world applications, and financial policies. The complexity of courses introduces variance between the predicted and actual results.

At the department level, discrepancies are often due to diversity of course content, heterogeneity of student populations, and subjectivity in grading. Departments like Medical Laboratory Science follows structured course content, consistent learning outcomes, and used standardized evaluations, this enable higher prediction accuracy. Conversely, departments such as Child and Family Studies involve qualitative assessment that are less predictable, resulting in lower model performance. In addition, departments with diverse student backgrounds or interdisciplinary content introduce greater variation in term GPA outcomes, and making prediction harder in these academic contexts.

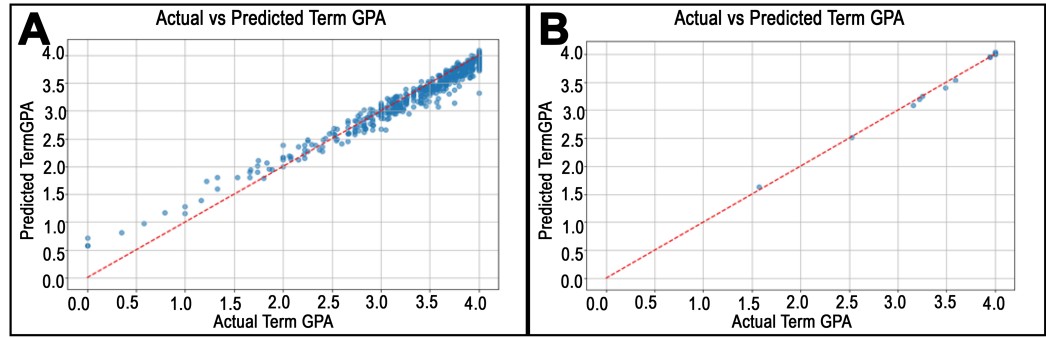

**Figure 6 Predicted *vs.* actual values plots for departments in College of Education.** (A) shows the highest RMSE in the Child and Family Studies Department. On the other hand, (B) shows the worst RMSE for the Information Media department.

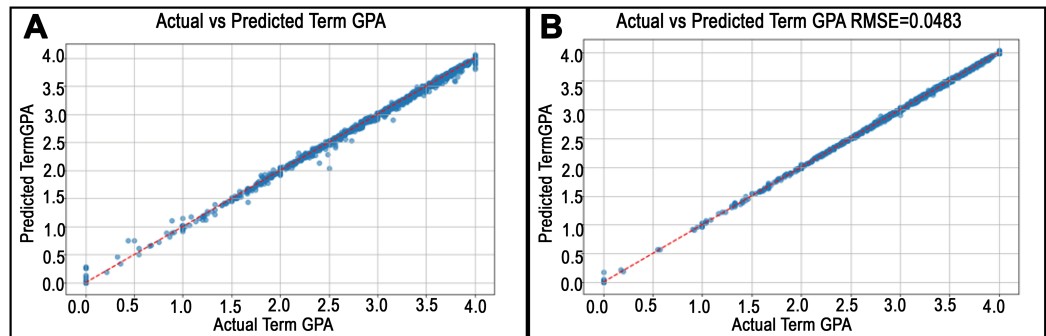

**Figure 7 Predicted *vs.* actual values plots for departments in College of Health.** (A) and (B) illustrate the best RMSE finding in the Kinesiology department and the worst RMSE in the department of Medical Laboratory Science, respectively, for the College of Health.

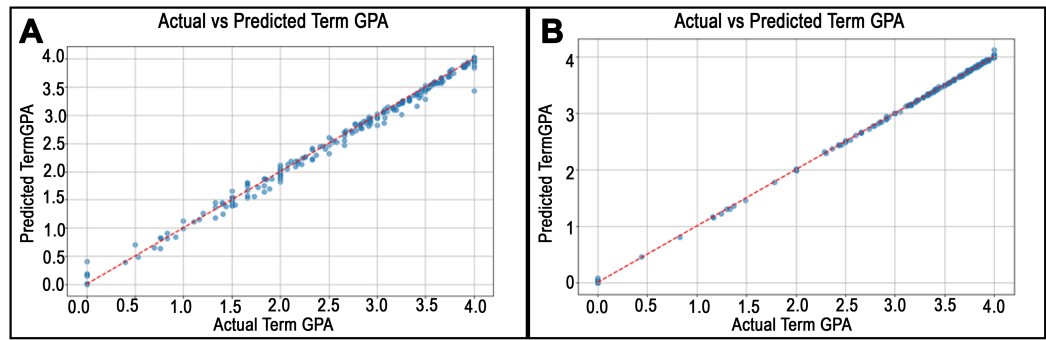

**Figure 8 Predicted *vs.* actual values plots for departments in College of Liberal Arts.** The best RMSE achieved in the College of Liberal Arts was in the Anthropology department, as shown in (A), while (B) shows the worst result in the Languages and Cultures department.

The plots shown in Fig. 6 through Fig. 10, illustrate the predicted *vs.* actual term GPA values for highest and lowest RMSE values of departments in each single college: College of Education and Learning Design, College of Health and Wellness Professions, College of

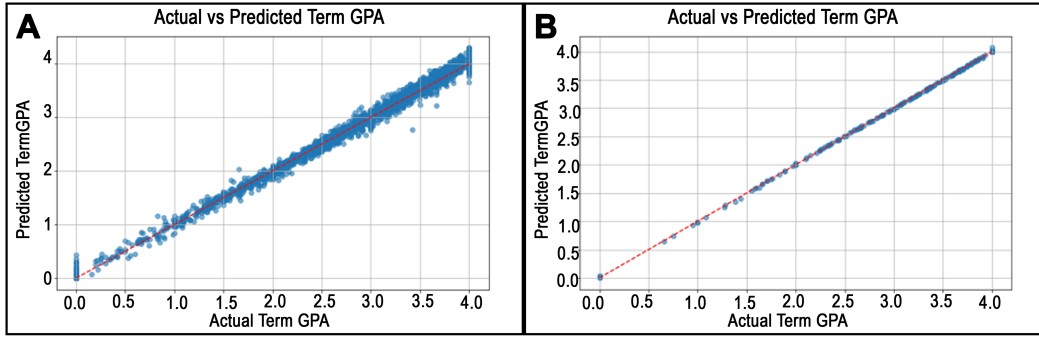

**Figure 9 Predicted *vs.* actual values plots for departments in College of Science and Engineering.** The best and the worst performance of the departments in the College of Science and Engineering is shown. (A) and (B) show the best and worst performing departments, the Biological Sciences and Physics and Astronomy departments.

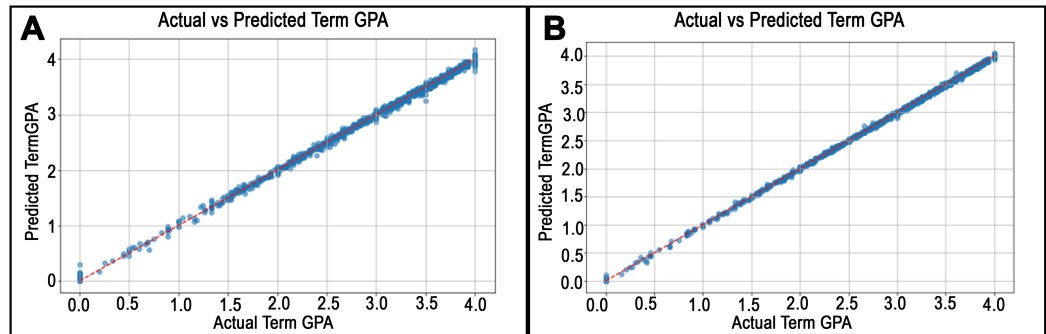

**Figure 10 Predicted *vs.* actual values plots for departments in Herberger Business College.** The Accounting department in (A) has the best RMSE, while the worst RMSE result for Information Systems is represented in (B).

Liberal Arts, College of Science and Engineering, and Herberger Business College. College of Education and Learning Design, the highest difference is reported in Child and Family Studies department is 14.6%, whereas the lowest difference is indicated as 3.1% in the department of Information Media. Similarly, in College of Liberal Arts, our findings indicate that there is difference between the actual and the predicted term GPA as 9.6% in Anthropology, and 1.4% in Languages and Cultures department as given in Fig. 8. On the other hand, there is slight differences among departments in Herberger Business College. The findings reveal homogeneity in prediction accuracy within the Herberger Business College, while greater variability is observed in the College of Education and Learning Design.

In summary, subjective evaluation provides variability and less quantifiable grading patterns which make lower prediction whereas objective evaluation introduces more quantifiable consistent data that improve the prediction accuracy. The proposed model will help advisors to observe the academic performance of their advisees through predicting the low-expectation students with failed or low term GPA. Consequently, this

**Table 10 A comparison with baseline ML classifiers and deep learning models.** Bold results means the highest and best results.

| Classifier | MAE | MSE | RMSE | $R^2$ |
|---|---|---|---|---|
| LR | 0.5283 | 0.4549 | 0.6745 | 0.5222 |
| KNN | 0.5509 | 0.5164 | 0.7186 | 0.4576 |
| DT | 0.6409 | 0.7561 | 0.8696 | 0.2058 |
| RF | 0.5053 | 0.4345 | 0.6592 | 0.5436 |
| SVR | 0.4987 | 0.4169 | 0.6457 | 0.5620 |
| RNN | 0.1074 | 0.0370 | 0.1925 | 0.9636 |
| CNN | 0.0720 | 0.0219 | 0.1479 | 0.9785 |
| Proposed model | **0.0059** | **0.0001** | **0.0108** | **0.9999** |

**Table 11 Comparing with related approaches.** Bold results means the highest and best results.

| Study | Model | Dataset | #Students | MAPE | MAE | MSE | RMSE |
|---|---|---|---|---|---|---|---|
| *Tsiakmaki et al. (2018)* | LR, SVM, DT, KNN | TEI of Western Greece | 592 | – | 1.21 | – | – |
| *Falát & Piscová (2022)* | LR, DT, RF | University of Zilina | 79 | 11.13 | – | 0.11 | – |
| *Obsie & Adem (2018)* | LR | Hawassa University | 134 | – | – | – | 0.0857 |
| *Elbadrawy et al. (2016)* | PLMR | University of Minnesota | 11,556 | – | – | – | 0.632 |
| *Prabowo et al. (2021)* | MLP-LSTM | Bina Nusantara University | 46,670 | – | 0.34 | 0.414 | – |
| *Alnomay, Alfadhly & Alqarni (2024)* | LR | King Saud University | 12,499 | – | 0.21 | – | – |
| *Akuma & Abakpa (2021)* | LR | Benue State University | 70 | – | 0.150 | 0.048 | 0.2199 |
| This study | LSTM | SCSU | 29,455 | **9.54** | **0.0059** | **0.001** | **0.0108** |

would be an effective strategy to avoid unexpected term GPA decline and improve the academic success.

## Validation of the proposed model

To validate our proposed approach, we compare our findings with five traditional ML classifiers, two deep learning models, including CNN and RNN, by using the same dataset, as indicated in Table 10, and the related work as presented in Table 11. The ML classifiers are linear regression (LR), K-nearest neighbor (KNN), decision tree (DT), random forest (RF), and support vector regressor (SVR). Among deep learning models, the LSTM had the lowest RMSE of 0.0108, indicating higher prediction accuracy. In contrast, the CNN and RNN models had RMSEs of 0.1479 and 0.1925, respectively. These findings reveal that the LSTM outperforms those two models in terms of capturing temporal relationships and complicated patterns in students academic records. LSTM is designed to detect both short-term and long-term dependencies in sequential data, making it ideal for predicting term GPA across multiple academic semesters. Unlike RNNs, which struggle with vanishing gradients, LSTMs use memory cells and gating mechanisms to preserve all relevant previous information in an effective fashion. CNNs are very efficient for spatial data, and less suited for time-series prediction. Also, compared with the related approaches, our proposed model achieves the highest prediction performance in all

reported evaluation metrics. For example, the findings in the study (*Prabowo et al., 2021*) introduced 0.34 and 0.414 for MAE and MSE. On the other hand, our results reveal 0.0059 and 0.001 for the same metrics.

Obviously, prediction results vary across universities due to differences in academic structure, assessment methods, students demographic, data availability, and how well prediction models align with institutional contexts. Universities use varying grading styles, some rely on objective assessments like exams and labs, while others use subjective evaluations such as projects (*Al-Ahmad et al., 2022*) and classroom engagement, leading to inconsistencies in GPA patterns. Additionally, factors like transfer status, course difficulty, and the absence of behavioral data influence model accuracy. As a result, even with similar predictive models, performance outcomes differ significantly depending on the educational environment and data characteristics of each institution.

The differences in performance of the proposed approach comparing to other universities stems from institutional, academic, and methodological variation. First, at SCSU, the dataset consists of 8 years and includes over 71,000 records from 29,455 students across five colleges and 46 departments. This provides diverse and comprehensive academic context. This contrasts with most of comparative studies that used small or homogeneous dataset that focused on one or two colleges. Second, the proposed model predicts term GPA at both of college and department levels, and detects deeper variation within the university that other related models may overlook. Third, the proposed model captures only academic and demographic features due to institutional constraints, so the behavioral and non-cognitive features which have been used by the previous related approaches may influence the prediction performance. Such reasons clarify why its performance may differ from the previous studies that conducted in more uniform academic environments.

## RESEARCH LIMITATIONS

Despite the promising results of the proposed model, this study presents some limitations. First, the model is designed to predict term GPA primarily for students in their third and fourth academic years, based on prior academic records. As a result, it may hard to generalize to newly students with limited historical data. Second, the study is restricted to data from a single institution, Saint Cloud State University, which limits the generalizability of the findings to other universities with differing curricular structures, grading systems, student populations, and assessment methodologies. Third, the dataset includes only academic and demographic features due to institutional data availability and privacy constraints. Consequently, the non-academic predictors such as psychological and socioeconomic features, motivation, and study habits were excluded. Additionally, behavioral data such as attendance, participation, or learning management system (LMS) interaction logs were not incorporated, which could have enriched temporal modeling of student performance. Fourth, course and instructor specific variables such as course difficulty, instructor grading factors, and class size were not included due to a lack of standardized data across departments. These factors may introduce noise or bias in the prediction process. Fifth, the current model architecture (LSTM) is optimized for short to

medium term sequence modeling. While this is suitable for the available data, other architectures (Transformers) could be explored in the future.

## CONCLUSION AND FUTURE WORK

Predicting term GPA is very important task to better evaluate the performance of students. This study introduces an analytical model that purposes to improve the prediction accuracy. The experiments conducted at college and department levels in Saint Cloud State University. With comparing to results of the related studies and ML classifiers, the findings reveal superior predictive performance with 9.54 MAPE, 0.0059 MAE, 0.0001 RMSE, and 99% $R^2$. These significant findings underline how the proposed model can increase the prediction performance of term GPA. Adapting such prediction model is effectively essential to help academic advisors detect students with low-expectation outcomes, and provide them with more sophisticated guidance strategies.

As future work directions, it will be worthwhile to predict the cumulative GPA. Obtaining such important information will help to compare the results with the current results. Furthermore, we plan to apply the proposed model on different datasets in other universities which would open the doors to compare the future findings with the current ones. It is interesting to apply different regression models, which it helps to discuss results with our current findings. Further, one important direction is to develop a tool based on this model to automate the prediction process. It is worth mentioning, we also plan to extend the model across multiple universities with varying academic styles, grading systems, and student populations. This will help to improve the generalizability and applicability of our model in a broader educational heterogeneous context. Finally, future work could be expanded to include collecting other significant features, such as non-cognitive, socioeconomic, course-specific contextual, and dynamic metrics of student engagement, and integrate all of them with the current dataset. Such a combination will help to show the learning behavior data of the students and instructors during the investigated academic terms, leading to make the proposed model a more reliable and generalized framework for identifying at-risk students. Such efforts have been left to be done in the future.

### Funding
The authors received no funding for this work.

### Competing Interests
The authors declare that they have no competing interests.

### Author Contributions
- Bilal I. Al-Ahmad conceived and designed the experiments, performed the experiments, analyzed the data, performed the computation work, prepared figures and/or tables, authored or reviewed drafts of the article, and approved the final draft.

- Abdullah Alzaqebah conceived and designed the experiments, performed the experiments, performed the computation work, prepared figures and/or tables, authored or reviewed drafts of the article, and approved the final draft.
- Rami Alkhawaldeh analyzed the data, performed the computation work, prepared figures and/or tables, authored or reviewed drafts of the article, and approved the final draft.
- Ala' M. Al-Zoubi conceived and designed the experiments, performed the experiments, performed the computation work, prepared figures and/or tables, and approved the final draft.
- Hsuehi Lo analyzed the data, performed the computation work, prepared figures and/or tables, and approved the final draft.
- Adel Ali analyzed the data, performed the computation work, authored or reviewed drafts of the article, and approved the final draft.

## Data Availability

The Saint Cloud State University (SCSU) Dataset (29,455 students over 8 years (2016–2024)) is available in the Supplemental File.

The code is available in the Supplemental File.

## Supplemental Information

Supplemental information for this article can be found online at http://dx.doi.org/10.7717/peerj-cs.3087#supplemental-information.

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
