# Peer review of "Predicting academic performance for students' university: case study from Saint Cloud State University"

_PeerJ Computer Science, doi:10.7717/peerj-cs.3087_

## Round 0.1 · original submission · Major Revisions

Please respond to the reviewers in your next revision.

Reviewer 1 ·

Basic reporting

Addressing these suggestions will significantly improve the manuscript's overall quality and increase its chances of meeting the necessary standards for publication.
Abstract: The explanation of the methodology lacks details on feature selection and model hyperparameter tuning.
1. Introduction.
• Clarify how the various prediction models compare in accuracy across different institutions.
• Explain the limitations of applying models to specific departments and how they may affect generalization.
• Expand on how non-cognitive factors influence GPA predictions, including potential data collection challenges.
• Strengthen the link between model performance and early identification of at-risk students for interventions.. Authors must read these to improve the introduction section: 1. A critical review of RNN and LSTM variants in hydrological time series predictions.

Experimental design

• Clarify how the "sequence length of 2" influences model accuracy and its role in time series.
• Specify the rationale behind choosing label encoding over one-hot encoding for categorical variables.
• Explain why MinMaxScaler was chosen and its impact on model performance compared to other scalers.
• Include more details on the LSTM architecture's hyperparameters, such as the number of epochs.

Validity of the findings

• Clarify the impact of student demographics and external factors on the feature importance calculations for better transparency.
• Provide more detailed explanations for discrepancies in prediction accuracy across different academic departments and colleges.
• Further justify the choice of experimental settings (e.g., specific model architecture) based on existing research.
• Expand on how modality and transfer student status are handled in the prediction model to improve robustness.

Additional comments

• Consider exploring how model generalization could be improved across diverse academic structures and grading systems.
• Expand future work by including validation across multiple universities to enhance model applicability and robustness.

·

Basic reporting

This manuscript presents an LSTM-based model for predicting students' semester GPAs. In general, this manuscript is well written and could be of interest to the knowledge base. However, there are some significant concerns about the presented methodology, analysis, and rationale that require a thorough revision.

1- Completeness of features: The study appears to omit several key characteristics that are widely known to have a significant impact on predicting student GPA. The manuscript should justify the exclusion of these characteristics:
Inherent course difficulty and historical student success rates in specific courses.
Dynamic metrics of student engagement (e.g., LMS activity, resource utilisation).
Course-specific contextual factors (e.g., instructor influence, modality, class size).
The student's actual course load (e.g., total number of credits) during the semester being predicted.
Relevant socioeconomic factors or significant outside commitments of the student. Not taking these variables into account may limit the explanatory power and generalisability of the model.

2- LSTM sensitivity and configuration: Although LSTMs are very powerful, they are sensitive to the characteristics of the input data and the prediction windows. The manuscript does not provide sufficient information in this regard.

3- Analysing the performance variations: The results show performance variations between different universities. The manuscript needs to provide a more in-depth analysis and explanation for these differences. Are they due to different student demographics, programme structures, data characteristics of each HEI, or limitations in the functional scope of the model? This analysis is critical to understanding the applicability of the model in different environments.

4- Evidence of model training: standard evidence of model stability and convergence is required, e.g., learning curves (with performance metrics such as loss or accuracy compared to training epochs) for both the training and validation sets. This will help to diagnose potential problems such as overfitting or insufficient training.

5- Visualisation of prediction accuracy: To better illustrate the performance of the model on this regression task, graphs should be created comparing the predicted GPA values with the actual GPA values for the test set. This is a common practice and helps to visualise the accuracy and error patterns (e.g., bias, variance) more intuitively than the aggregated metrics alone.
Rationale for model selection: The manuscript should include a clear justification for the choice of LSTM over other potentially suitable or more modern time series AI architectures. In particular, a discussion comparing LSTM to transformer-based models, known for their effectiveness in capturing long-range dependencies, would strengthen the paper by justifying why LSTM was considered best suited for this specific problem and dataset.

Rigorous benchmarking: Comparison with baseline models needs to be carefully considered as these models are subject to limitations ("outdated publications, use of models with low depth of learning, or use of fewer samples"). Comparing the proposed model with potentially weak or poorly implemented baseline models may distort the comparison and inflate the supposed superiority of the proposed method, especially if it has been evaluated with other data sets or under different conditions. Claims of superior results need to be supported by comparisons with meaningful, relevant base models that are rigorously and fairly evaluated, ideally with the same dataset.

Experimental design

-

Validity of the findings

-

---

## Round 0.2 · Minor Revisions

I appreciate the significant edits made by the authors in the first round of reviews. I have a few other suggestions before acceptance:
- I believe the work requires IRB approval (or at least an IRB waiver). This should be specified in the paper before acceptance
- I suggest expanding the research limitations paragraph to be more holistic
- There are quite a few places to clean up the writing. Example: Sentences like “There are different educational factors have been studies…” should be revised.
- Use consistent terminology—e.g., “term GPA,” “semester GPA,” and “GPA” appear interchangeably

Reviewer 1 ·

Basic reporting

Accepted

Experimental design

Accepted

Validity of the findings

Accepted

Additional comments

Accepted

---

## Round 0.3 · accepted · Accept

Thank you for addressing all of the reviewer comments and concerns. I have assessed this version myself and I am happy with it. It is ready for publication.